# ADVERSARIAL CO-EVOLUTION OF LLM-GENERATED POLICIES AND ENVIRONMENTS VIA TWO-PLAYER ZERO-SUM GAME

## ABSTRACT

A central challenge towards building agents that continually improve is that training environments are typically fixed or manually designed. This restricts continual learning and generalization beyond the training distribution. We address this with `Covolve`, a co-evolutionary framework that leverages large language models (LLMs) to generate both environments and agent policies, expressed as executable Python code. We model the interaction between environment and policy designers as a two-player zero-sum game, ensuring adversarial co-evolution in which environments expose policy weaknesses and policies adapt in response. To guarantee robustness and prevent forgetting, we compute the mixed strategy Nash equilibrium (MSNE) of this game, yielding a meta-policy that remains robust across all generated environments rather than overfitting to the most recent one. This process induces an automated curriculum in which environments and policies co-evolve toward increasing complexity. Experiments in urban driving, maze-solving, and 2D navigation showcase that `Covolve` produces progressively more complex environments. The MSNE meta-policy also ensures that the agent does not forget to solve previously seen environments, all the while learning to solve unseen ones. These results demonstrate the potential of LLM-driven co-evolution to achieve open-ended learning without predefined task distributions or manual intervention.

## 1 INTRODUCTION

Developing agents that continually acquire new skills in dynamic, unpredictable settings remains a core challenge in AI. Most current training pipelines still rely heavily on large amounts of human-curated data, which is both costly and results in agents with limited generalization capabilities (Villalobos et al., 2024). While reinforcement learning (RL) offers a promising alternative by enabling agents to learn through potentially unlimited interactions within a simulator (Silver and Sutton, 2025), it still suffers from a critical bottleneck in that the environments, in which agents are trained, are either fixed and/or manually designed. Due to the difficulty of designing a distribution of training environments that captures the full range of real-world variability (Clune, 2020), RL agents often fail to generalize beyond the narrow distribution of environments encountered during training (Ghosh et al., 2021; Kirk et al., 2023; Korkmaz, 2024). Achieving robustness and transferability requires exposing agents to a diverse, evolving curriculum of environments that adapt to their growing capabilities and that continually acquire new behaviors (Cobbe et al., 2020; Korkmaz, 2024).

*Unsupervised Environment Design* (UED) (Dennis et al., 2020; Jiang et al., 2021) addresses these limitations in training environments by automatically generating a *curriculum of environments* that adapts in difficulty based on the agent's performance. By dynamically tailoring environments to expose and challenge the agent's weaknesses, UED encourages continual learning. However, UED typically generates environments via randomization or simple heuristics – rather than intelligently – limiting diversity and relevance. We overcome this by introducing `Covolve`, a co-evolutionary framework that frames UED as a two-player zero-sum game and leverages LLM-based code generation with rich priors to *intelligently* design both environments and policies. In `Covolve`, an LLM-powered environment designer and policy designer compete adversarially to co-evolve more challenging *levels* and more capable policies, respectively (see Figure 1). Concretely, we make the following contributions:

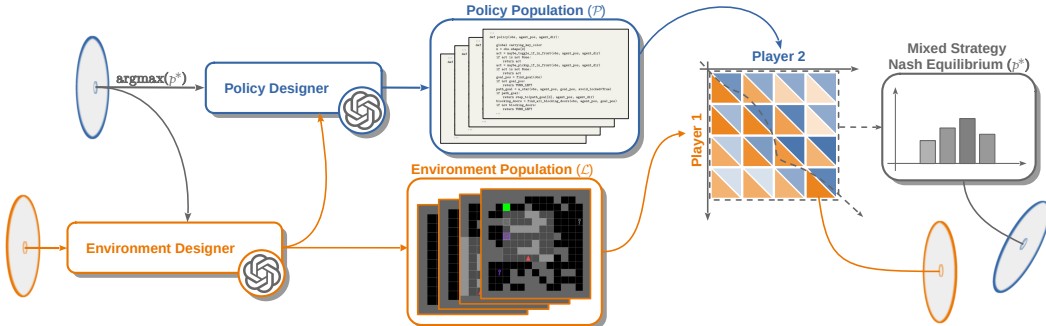

Figure 1: An overview of the proposed `Covolve`, comprised of an Environment Designer and a Policy Designer that co-evolve by playing a two-player zero-sum game. The Environment Designer generates increasingly challenging environments (as code), while the Policy Designer creates policies (as code) to solve them. A mixed-strategy Nash equilibrium enables robust, open-ended learning through continual adaptation.

**(1) LLM-Driven Co-Evolution.**    Previous research has explored LLM-driven environment (Faldor et al., 2025) and policy design (Liang et al., 2023) through *code-based outputs*, where both environments and policies are represented as code. This offers several key advantages over neural representations: (i) generalizability to unseen scenarios (Inala et al., 2020; Trivedi et al., 2021), (ii) modularity and reuse of code (Wang et al., 2024a; Ellis et al., 2023), (iii) verifiability and interpretability (Bastani et al., 2018; Verma et al., 2018), (iv) expressiveness through Python's Turing-completeness enabling representation of any computable environment or policy, and (v) rich priors from LLMs providing strong commonsense knowledge for automatic generation of diverse tasks without hand-crafted templates (Faldor et al., 2025). However, existing frameworks typically focus on either environment generation without fostering robust agents (Faldor et al., 2025), or on policy design without continual adaptation (Liang et al., 2024). In contrast, `Covolve` closes the loop by leveraging LLM-driven co-evolution of both agents and environments, automatically generating an adaptive curriculum that promotes continual learning.

**(2) Game-theoretic Framework for Robust Policy Design.**    We frame the co-evolution as a two-player zero-sum game between a policy player and an environment player. At each iteration, `Covolve` maintains populations of policies and environments, constructs a meta-game by evaluating all pairs, and computes the Nash equilibrium. This yields a robust meta-policy that optimally handles worst-case scenarios, while generating environments that effectively challenge current policies (Lanctot et al., 2017). The Nash equilibrium provides theoretical guarantees (Nash, 1951), balancing training difficulty and ensuring strong performance across seen and unseen environments. In contrast, previous works (Wang et al., 2019) only optimize for independent policies for each environment.

**(3) Empirical Evidence of Emergent Curriculum and Generalization.**    We empirically demonstrate that `Covolve` produces increasingly challenging environments across diverse domains (urban driving, maze-solving, and 2D navigation), with generated levels exhibiting escalating complexity and diversity over time. Crucially, our evaluation shows that computing the MSNE is essential to prevent catastrophic forgetting, unlike approaches like Eurekaverse (Liang et al., 2024) which retain only the latest best policy and finetune on new environments, leading to forgetting.

## 2    PRELIMINARIES

### 2.1    UNSUPERVISED ENVIRONMENT DESIGN (UED)

Formally, a UED is defined over an *underspecified partially observable Markov decision process* (UPOMDP) (Dennis et al., 2020), given by the 8-tuple $\mathcal{M} = (\Theta, \mathcal{S}, \mathcal{A}, \mathcal{O}, T, O, R, \gamma)$, with the last seven elements having the same meaning as in a standard POMDP: $\mathcal{S}$, $\mathcal{A}$, and $\mathcal{O}$ are the sets of states,

actions, and observations, respectively, $T$ and $O$ denote the transition and observation functions, $R$ is the reward function, and $\gamma \in [0, 1]$ the discount factor.

The first element $\Theta$ of a UPOMDP $\mathcal{M}$ denotes the space of *underspecified* environment parameters (e.g., number and position of obstacles, size of the grid). Picking a specific $\theta \in \Theta$ materializes a concrete POMDP. A UPOMDP can hence be viewed as a set of POMDPs. A concrete set of parameters $\theta \in \Theta$ is also referred to as a *level*. The choice of $\theta$ may influence the reward function $R : \mathcal{S} \times \mathcal{A} \times \Theta \to \mathbb{R}$, the transition function $T : \mathcal{S} \times \mathcal{A} \times \Theta \to \Delta(\mathcal{S})$, and the observation function $O : \mathcal{S} \times \Theta \to \mathcal{O}$, where $\Delta(S)$ is the set of all probability distributions over $\mathcal{S}$.

Given a level $\theta \in \Theta$, the expected discounted return (i.e. utility) of a policy $\pi$ and a level $\theta$ is denoted as $U_\theta(\pi) = \mathbb{E}_{\tau \sim (\pi, \theta)}[G_\tau]$, with $\tau$ denoting trajectories sampled under the policy $\pi$ at level $\theta$. $G_\tau$ is the sum of discounted rewards for each trajectory: $\sum_{t=0}^{T} \gamma^t r_t$, with $r_t$ being the reward collected at time step $t$. The optimal policy for level $\theta$ is then given by $\pi_\theta^\star = \arg\max_\pi U_\theta(\pi)$.

The goal of UED is to train a policy that performs well across a broad distribution of environments. To this end, UED is typically framed as a two-player game, with an adversary $\Lambda$ from which we can sample levels given a policy: $\theta \sim \Lambda(\pi)$. The adversary's goal is to identify levels that challenge the policy $\pi$ by optimizing the utility function $U_\theta(\pi)$ that exposes its weaknesses.

A simple example is that of a *minimax adversary* (Morimoto and Doya, 2005; Pinto et al., 2017; Samvelyan et al., 2023), which is a point distribution $\Lambda(\pi) = \arg\min_{\theta \in \Theta} U_\theta(\pi)$ and proposes new levels to minimize the policy performance. In response, a *maximin* policy is one that tries to perform well under the most adversarial level $\pi^* = \arg\max_{\pi \in \Pi} \min_{\theta \in \Theta} U_\theta(\pi)$. However, solving this exactly is computationally intractable. In the following section, we introduce an efficient approximation method.

## 2.2 POLICY SPACE RESPONSE ORACLES (PSRO)

PSRO (Lanctot et al., 2017) is a general framework for multi-agent learning that addresses the fundamental challenge of non-stationarity in multi-agent environments. In such settings, the optimal policy for one agent depends on the policies of other agents, creating a moving target that makes traditional single-agent reinforcement learning approaches ineffective. Rather than attempting to learn a single "best" policy, PSRO builds and maintains a diverse population of policies over time. This approach provides robustness against various opponent strategies and reduces exploitability in competitive scenarios.

In a 2-player setting, the PSRO framework operates through the following four-step iterative process: (1) Each player $i \in \{1, 2\}$ maintains a growing set of policies $\mathcal{P}^i = \{\pi_1^i, \pi_2^i, \ldots, \pi_t^i\}$, creating a library of strategies for that player. (2) PSRO constructs a payoff matrix $\mathbf{M} \in \mathbb{R}^{|\mathcal{P}^1| \times |\mathcal{P}^2|}$ by evaluating all pairwise policy combinations, where each entry represents the expected payoffs for players 1 when player 1 uses policy $\pi_i^1$ and player 2 uses policy $\pi_j^2$. (3) The framework computes a meta-policy for player 1 that determines how to mix the existing policies in the population. (4) For player 2, a new *best response* policy $\pi_{t+1}^2$ is trained to maximize performance against the player 1 meta-policy, and is subsequently added to the policy population $\mathcal{P}^2$ of player 2. This iterative process continues until convergence, resulting in a diverse and robust policy population.

## 3 METHODOLOGY

We adapt PSRO to UED by applying it as a two-player game between an **Environment Designer**, which generates new levels to challenge the meta-policy, and a **Policy Designer**, which synthesizes new policies to solve them. Given a *minimax* objective, this interaction forms a two-player zero-sum game, as summarized in Algorithm 1, with additional algorithmic details presented in Appendix A. Since both environments and policies are represented as executable Python code, Figure 2 illustrates their co-evolution across iterations.

**Policy Designer.** For each newly proposed level, the Policy Designer $\Psi$ synthesizes and iteratively refines the policy to adapt to the level. For a given level $\theta$, the policy designer generates $K$ candidate policies given as $\{\tilde{\pi}_1, \ldots, \tilde{\pi}_K\} = \Psi_{\text{LLM}}(\mathcal{O}^\theta, \mathcal{A}^\theta)$. Here, $\Psi_{\text{LLM}}$ denotes the LLM-based policy designer that uses the observation space and the action space from the level $\theta$ to design policies

---

**Algorithm 1** `Covolve`

---

1: **Require:** Initial environment $\theta_0$
2: **Hyperparameters:** Total iterations $T$, candidate generated per level $K$
3: Initialize environment levels $\mathcal{L} \leftarrow \{\theta_0\}$, policy sequence $\mathcal{P} \leftarrow (\ )$, Payoff Matrix $\mathbf{M} = [\ ]$
4: **for** $t = 0$ to $T$ **do**
5:     `#1. Policy Design`
6:     Generate $K$ candidates: $\{\tilde{\pi}_1, \ldots, \tilde{\pi}_K\} = \Psi_{\text{LLM}}(\mathcal{O}^{\theta_t}, \mathcal{A}^{\theta_t})$
7:     $\pi_t = \arg\max_j U_\theta(\tilde{\pi}_j)$
8:     Append $\mathcal{P} \leftarrow \pi_t$
9:     `#2. Update Payoff Matrix` $\mathbf{M}$
10:     **for** $i, j = 0$ to $t$ **do**
11:         $m_{ij} \leftarrow U_{\theta_j}(\pi_i)$
12:     **end for**
13:     `#3. Recompute MSNE`
14:     $\pi_{\text{MSNE}} \leftarrow \text{SolveNash}(M)$                                       ▷ Eq. 1
15:     `#4. Best Response Environment Design`
16:     Generate $K$ candidates: $\{\tilde{\theta}_1, \ldots, \tilde{\theta}_K\} = \Lambda_{\text{LLM}}(\pi_{\text{MSNE}})$
17:     $\theta_{t+1} = \arg\min_j\{U_{\tilde{\theta}_j}(\pi_{\text{MSNE}})\}$
18:     Add $\mathcal{L} \leftarrow \theta_{t+1}$
19: **end for**
20: **Return:** MSNE Policy $\pi_{\text{MSNE}}$, environment levels $\mathcal{L}$

---

as code. It further uses (in-context) inspirations from the previous policies. The candidate $\pi = \arg\max_j\{U_\theta(\tilde{\pi}_j)\}$ that maximizes the performance on $\theta$ is selected[1] added to a growing sequence of policies $\mathcal{P}$.

While each policy is tailored to an individual level, our broader goal is to obtain an approximation of the optimal policy $\pi^*$ that performs well across levels. Since the interaction between the policy designer and the environment designer forms a two-player zero-sum game, a mixed-strategy Nash Equilibrium (MSNE) provides a principled solution, ensuring that the obtained policy is robust under adversarial conditions. Building on PSRO, we achieve this by maintaining a growing sequence of policies $\mathcal{P} = \{\pi_1, \ldots, \pi_r\}$ and previously generated levels $\mathcal{L} = \{\theta_1, \ldots, \theta_r\}$, where each policy $\pi_i$ is optimized for its corresponding level $\theta_i$ using the process described above. Let the payoff matrix be $\mathbf{M} \in \mathbb{R}^{r \times r}$, where each entry $m_{ij} = U_{\theta_j}(\pi_i)$ denotes the expected return of policy $\pi_i$ on level $\theta_j$. The *new* MSNE is then computed via a minimax optimization where the policy agent maximizes its worst-case expected payoff (Osborne and Rubinstein, 1994):

$$p^\star = \arg\max_{p \in \Delta^r} \min_{j \in \{1, \ldots, r\}} \sum_{i=1}^{r} p_i m_{ij} \quad \text{where,} \quad \Delta^r = \left\{ p \in \mathbb{R}^r \ \middle| \ \sum_{i=1}^{r} p_i = 1, \ p_i \geq 0 \ \forall i \right\} \quad (1)$$

Here, $p^\star$ is the mixture weights over policies that maximizes the worst-case expected return across all levels, defining the MSNE policy $\pi_{\text{MSNE}}$, where each policy $\pi_i$ is sampled from $\pi_{\text{MSNE}}$ with probability $p_i^*$. At the start of each episode, the agent samples a policy $\pi \sim \pi_{\text{MSNE}}$ and takes actions according to $a \sim \pi$ in the given level.

**Environment Designer.** The Environment Designer generates a new level as a *best response* to the current MSNE policy. Its goal is to minimize the expected return of the mixture policy, thereby revealing its weaknesses. This adversarial loop encourages curriculum-like progression, automatically increasing environment difficulty in response to agent improvement.

Following a similar evolutionary strategy, the environment designer uses an LLM-based adversary $\Lambda_{\text{LLM}}$ to generate $K$ candidate mutations of the current environment $\{\tilde{\theta}_1, \ldots, \tilde{\theta}_K\} = \Lambda_{\text{LLM}}(\pi_{\text{MSNE}})$. The candidate that minimizes performance under $\pi_{\text{MSNE}}$ is selected.

---

[1]The best policy can be further evolved using the performance utility $U_\theta(\tilde{\pi}_i)$ as feedback – analogous to training a neural network policy on the level $\theta$, yielding a final policy optimized for that level.

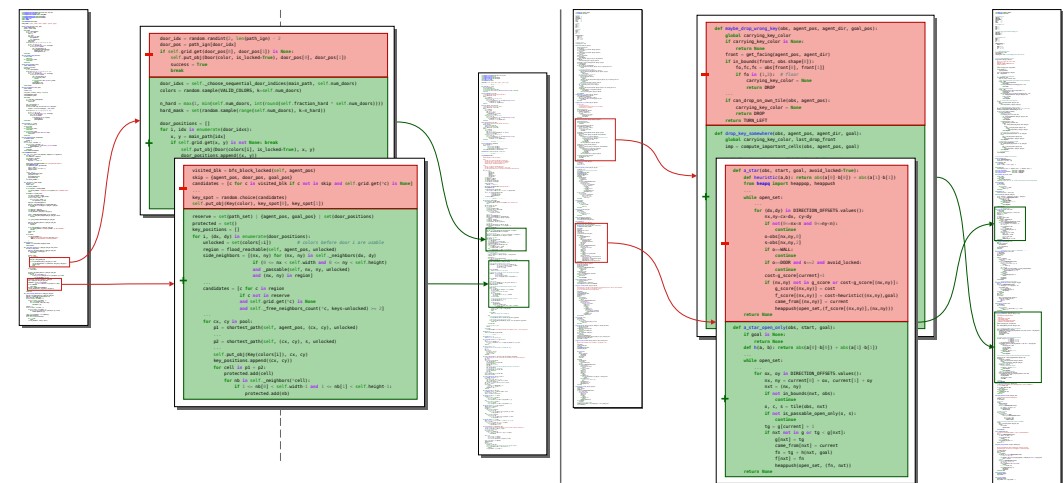

Figure 2: Illustration of how the Environment Designer (*left*) and the Policy Designer (*right*) are evolving both the complexity of the environment, as well as the complexity of the policy. Highlighted code blocks (*left*) illustrate how the environment evolves from random generation to a structured, parameter-driven design, while simultaneously guaranteeing solvability and tunable chokepoints. Highlighted code blocks (*right*) illustrate how the policy, meanwhile, evolves by adding better object handling (keys and doors in this example), while simultaneously improving the used A* pathfinder algorithm for more deliberate navigation.

The best-performing environment is then added to the level set $\mathcal{L}$, and a new policy is synthesized in response. This new policy is added to the policy sequence $\mathcal{P}$, and the cycle repeats, continuously enriching both policy and environment sequences.

## 4 RELATED WORKS

**Unsupervised Environment Design (UED).** Domain randomization (DR) exposes agents to a broad distribution of environments (Jakobi, 1997; Tobin et al., 2017) but lacks adaptivity and often produces trivial or unsolvable tasks (Dennis et al., 2020). UED addresses this by automatically generating curricula tailored to agent performance. For instance, minimax adversarial training selects environments that minimize the agent's reward (Morimoto and Doya, 2005; Pinto et al., 2017; Samvelyan et al., 2023). However, it can produce overly difficult tasks unless constrained (Dennis et al., 2020). Regret-based methods like PAIRED (Dennis et al., 2020) address this by defining regret relative to an approximated optimal policy to ensure solvability. While our work uses a minimax adversary, future directions could incorporate regret-based strategies to avoid generating unsolvable levels. Crucially, our LLM-driven co-evolution introduces intelligent, data-driven priors that enable the design of more challenging and relevant environments than classical, heuristic-based UED.

**LLMs for Environment Design.** Recent work uses code-generating LLMs to automate environment design (Faldor et al., 2025; Wang et al., 2024b), world model generation (Tang et al., 2024; Dainese et al., 2024), and reward specification in RL (Hazra et al., 2025; Ma et al., 2024). However, most frameworks either decouple environment and agent learning or focus only on environment generation, limiting agent robustness. Our framework enables fully closed-loop co-evolution, automatically generating a curriculum that adapts to both agent and environment. Closest to our approach is Eurekaverse (Liang et al., 2024), which also co-evolves agents and environments but retains and iterates with only the latest best policy. This leads to reduced robustness and agents forgetting to solve previously seen environments. In contrast, we use game-theoretic principles to maintain a diverse policy population via a mixed-strategy Nash equilibrium, enhancing generalization and stability.

**LLMs for Policy Design.** Parallel to environment design, LLMs have been used to synthesize modular, generalizable, and interpretable code-based policies. Approaches like Code-as-Policies(Liang et al., 2023), RL-GPT(Liu et al., 2024), and ProgPrompt(Singh et al., 2022) leverage LLMs to generate executable plans or combine code with RL controllers, but are typically limited to narrow

task distributions. In contrast, our approach constructs robust and continually adaptive policies that learn within an open-ended, co-evolving curriculum.

**Self-Play in LLMs.** Our work is also related to the paradigm of self-play, where models take on dual roles to create a self-improvement loop. Herein, LLMs create copies of each other with different roles to improve without human data reliance. This has been used in domains like coding (Coder-Tester Agents) (Wang et al., 2025; Lin et al., 2025) and reasoning (Challenger-Solver Agents) (Huang et al., 2025; Chen et al., 2025). The improvement step is directly applied to the LLMs, which can be inefficient for domains where solutions can be represented by compact policies rather than large, monolithic models. In contrast, `Covolve` harnesses LLMs to drive the design of specialized agents that are modular, interpretable, and easier to deploy. A more concurrent work is Bachrach et al. (2025) where LLMs produce strategies as code for playing against a Nash equilibrium mixture over the current population of strategies.

## 5 EXPERIMENTS

We evaluate `Covolve` across three complementary domains that capture distinct challenges in agent learning. MiniGrid (Chevalier-Boisvert et al., 2023) requires symbolic planning in procedurally generated mazes with sequential dependencies such as keys and doors. PyGame (PyGame Community, 2000–2024) emphasizes continuous 2D navigation with geometric reasoning and collision constraints, with difficulty scaling through denser obstacles and narrower traversable passages. CARLA (Dosovitskiy et al., 2017) provides a high-fidelity urban driving setting with partial observability, dynamic vehicles, pedestrians, and traffic lights. Together, these domains encompass planning, geometric navigation, and realistic multi-agent driving, forming a principled testbed for evaluating curriculum emergence and robustness under co-evolution.

### 5.1 ENVIRONMENTS AND TASKS

All environments and policies are generated with GPT-4.1 (OpenAI, 2024). Prompts are in Appendix C. All code was dynamically validated and executed with `exec()`. For each policy–environment pair, we evaluated the payoff $U_\theta(\pi) \in [0, 1]$ averaged over 100 episodes to populate the payoff matrix $\mathbf{M}$. Complete environment specifications (action/observation spaces, task scaling, termination, and feasibility checks) are provided in Appendix B.

**MiniGrid Maze Solving.** We use the `MiniGrid-Empty` environment as a base for generating symbolic maze-solving tasks. It is a fully-observable environment with $n \times n \times 3$ grids, augmented with the agent's absolute position and orientation, resulting in an $n \times n \times 3 + 2$ observation vector, where $n$ is the grid width and height. Each cell encodes object type, color identifier, and state (e.g., open/closed doors). The agent acts in a 5-action discrete space (*turn, move, pick up, drop, toggle*). Difficulty is scaled by enlarging grids, adding walls, and introducing locked doors with keys that must be retrieved and used in sequence, enforcing multi-step planning even in small mazes. We use handcrafted heuristics to validate if the generated environments are feasible (c.f., § 6 Limitations). Episodes terminate when the agent reaches the goal tile or when the step horizon is reached. A selection of evolved environments is shown in Figure 3, with further implementation details in Appendix B.1.

**PyGame 2D Navigation.** To test LLM's ability to deal with *continuous action spaces*, we use a custom 2D navigation environment in which a circular agent must reach a rectangular goal zone while avoiding fixed rectangular obstacles. States are fully observable, consisting of the agent's position and a list of all objects (obstacles and goals) with their positions and sizes (growing in size for each new level). The agent acts in a continuous space through 2D velocity commands. Difficulty increases when obstacles are added, the agent–goal distance is increased, and narrow passages are created that may block traversal. This requires agents to identify traversable corridors relative to their size and to plan long detours when direct routes are infeasible. Episodes terminate when the agent overlaps the goal zone or when the step horizon is reached. A selection of evolved environments is shown in Figure 4, with further details in Appendix B.2.

**CARLA Urban Driving.** We evaluate urban driving in `CARLA Town01`, a high-fidelity simulator with vehicles, pedestrians, and traffic lights. The vehicle follows a prescribed route using continuous

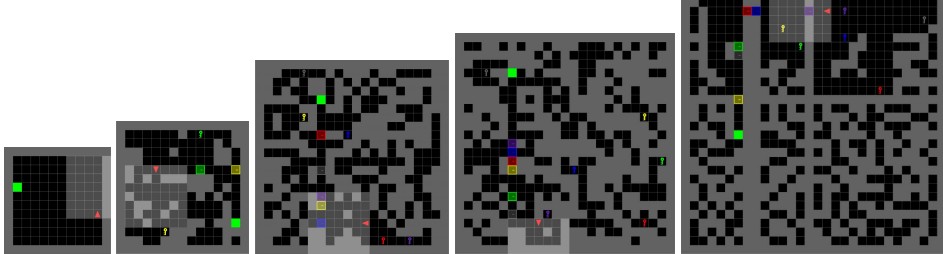

Figure 3: A selection of evolved MiniGrid environments produced by `Covolve`. Complexity increases from empty grids to larger mazes with dense walls and locked doors requiring corresponding keys. The agent must reach the green goal tile, often by planning multi-step sequences of key retrieval and door unlocking.

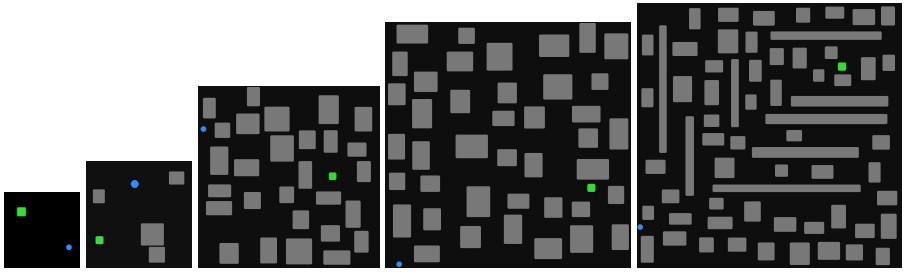

Figure 4: A selection of evolved PyGame environments produced by `Covolve`. Tasks progress from open arenas to cluttered maps with dense obstacles and narrow corridors. The agent must reach the rectangular goal zone while navigating collision-free paths through increasingly constrained layouts.

steering, throttle, and brake controls. Observations are egocentric and partial, consisting of the vehicle's kinematics, the nearest traffic light, and compact features of nearby vehicles and pedestrians. Task difficulty increases with varying traffic density and pedestrian activity, introducing adversarial behaviors such as abrupt braking or traffic-light violations. Episodes terminate upon route completion (success) or any infraction (collision, red-light violation, or timeout). This setting tests policy robustness under partial observability and multi-agent interactions with stochastic and sometimes adversarial actors. A selection of evolved environments is shown in Figure 5, with further details in Appendix B.3.

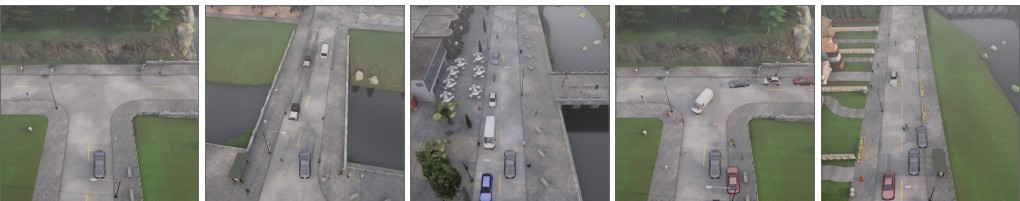

Figure 5: Selected CARLA environments produced by `Covolve`. Tasks progress from urban driving on empty roads to crowded streets with increasingly aggressive actor behaviors. The task for the agent is to drive along the street while following traffic rules (such as stopping at red lights), and at the same time, adjust to increasingly unpredictable behaviors of fellow drivers and pedestrians.

## 5.2 RESULTS

We use *Eurekaverse* (Liang et al., 2024) as our baseline, which iteratively trains multiple policies per level, retains the best one, generates a harder level, and initializes the next policy with the best policy's weights. For fair comparison, we represent Eurekaverse policies as Python code rather than neural networks. Unlike our method, Eurekaverse advances difficulty using only the best policy (not MSNE), which can lead to catastrophic forgetting. Because co-evolution produces distinct

environment archives across runs, results from different seeds are not directly comparable: averaging would mix scores over non-identical evaluation sets. We therefore report results from a single run.

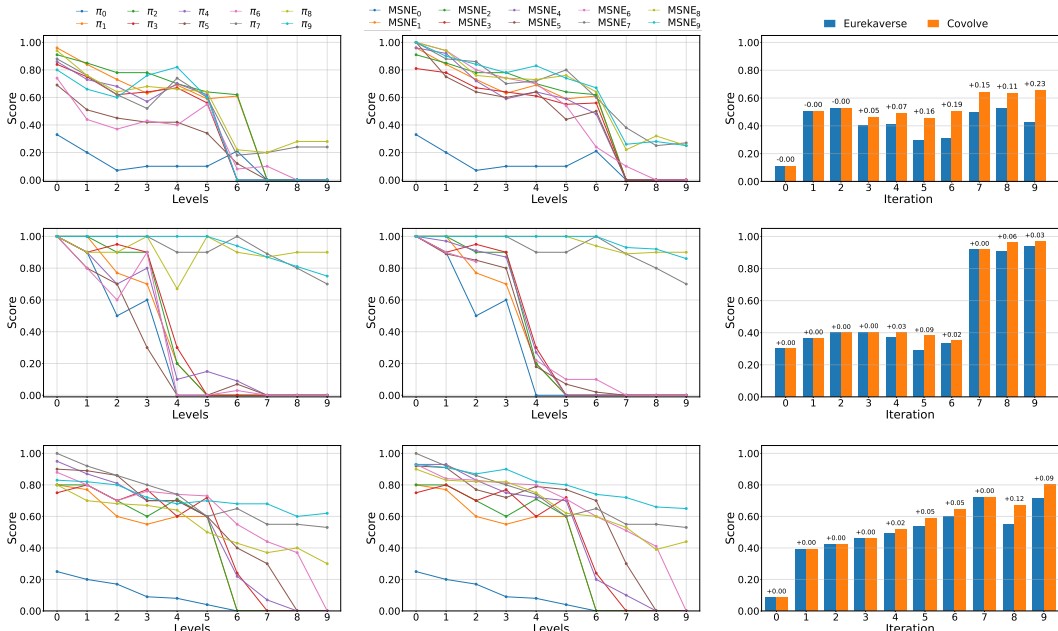

Figure 6: Eurekaverse vs. `Covolve` results on MiniGrid maze-solving (*top*), PyGame 2D navigation (*middle*), and CARLA urban driving (*bottom*). *Left:* Eurekaverse policies across levels, *center:* `Covolve` (MSNE) policies across levels, and *right:* direct comparison of the $k^{th}$ Eurekaverse and `Covolve` policies averaged across all levels.

The results, seen in Figure 6, report three perspectives: (i) Eurekaverse policy $\pi$ evaluated on all levels, (ii) the corresponding `Covolve` (MSNE) policies across all levels, and (iii) the direct comparison of Eurekaverse policy $\pi$ and `Covolve` policy at iteration $k$ averaged across all levels. The results show that Eurekaverse policies specialize to the environments for which they were tailored, and thereby fail to perform well across previous levels (Figure 6: *left*). The MSNE mixture, on the other hand, preserves competence across all levels and improves over time (Figure 6: *center*). A direct comparison between a single (latest) policy strategy and an MSNE strategy verifies that MSNE consistently outperforms the latest policy when the equilibrium is non-trivial (Figure 6: *right*).

## 5.3 GENERALIZATION

We test whether policies transfer beyond co-evolution by evaluating on *unseen standardized environments*: `DoorKey-16x16-v0` (MiniGrid), `LockedRoom-v0` (MiniGrid), and CARLA `Town02`. At iteration $k$, we compare an Eurekaverse-style latest-only policy with our `Covolve` MSNE mixture under identical rollout settings. The resulting success rates are reported in Table 1, while the rationale for these environments and their differences from the evolved tasks are detailed in Appendix D.1.

| Environment | Eurekaverse | Covolve |
|---|---|---|
| `DoorKey-16x16-v0` (MiniGrid) | 0.86±0.06 | **0.95±0.02** |
| `LockedRoom-v0` (MiniGrid) | 0.47±0.03 | **0.56±0.03** |
| `Town02` (CARLA) | 0.30±0.03 | **0.40±0.02** |

Table 1: Results on generalization to unseen environments. Success rates (mean±std) on standardized MiniGrid (`DoorKey-16x16-v0`, `LockedRoom-v0`) and CARLA (`Town02`) benchmarks, comparing Eurekaverse and `Covolve`.

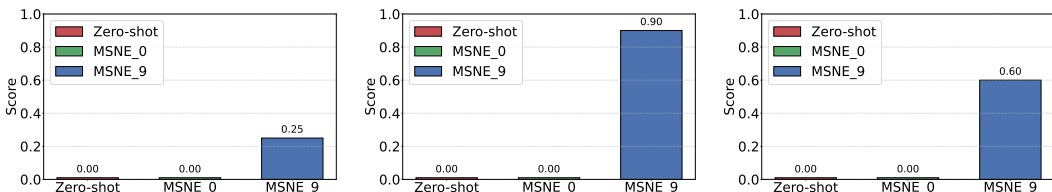

Figure 7: Result on curriculum learning. Direct training on the hardest environment ("Zero-shot") fails, while co-evolutionary MSNE with curriculum (*right bars*) yields non-trivial performance.

## 5.4 ABLATION STUDIES

**Is curriculum necessary?**   To assess the importance of curriculum, we have the LLM generate policies for the hardest environment in each domain (MiniGrid, PyGame, CARLA) in a zero-shot manner, selecting the best policy@$k$. This results in near-zero success, while co-evolution with MSNE leads to steady improvement. This demonstrates that progressive curriculum design is essential, as seen in Figure 7.

**How do neural network-based RL approaches compare to LLM-generated code-as-policies in solving fully observable environments?**   To test whether standard RL can solve our fully observable evaluation tasks, we trained representative Stable-Baselines3 agents (Raffin et al., 2021b) – PPO (Schulman et al., 2017) and SAC (continuous control) (Haarnoja et al., 2018) for PyGame, and PPO and QR-DQN (discrete) (Dabney et al., 2017) for MiniGrid. Despite full observability, performance degrades sharply with task complexity: PPO and QR-DQN achieve near-zero reward and success on the harder environments (SAC shows only limited improvements). See Appendix D.2 for details regarding training settings, curves, and success rates.

## 6 CONCLUSIONS

`Covolve` casts environment–policy co-evolution as a two-player zero-sum game and selects controllers via a mixed-strategy Nash equilibrium (MSNE) on the empirical policy–environment payoff matrix. On the same archive of co-evolved levels, the MSNE meta-policy maintains performance on earlier levels, whereas the latest-only selection (Eurekaverse) does not retain it uniformly as the archive grows. On the testing environments (i.e., `DoorKey-16x16`, `LockedRoom`, and `Town02`) with identical rollout settings, `Covolve` outperforms Eurekaverse. These results demonstrate that combining LLM-based generation with game-theoretic selection can sustain learning across evolving tasks while improving transfer to unseen ones.

### 6.1 LIMITATIONS & FUTURE WORK

LLM-generated environments can sometimes be infeasible – for example, producing unsolvable mazes – which may require intervention through handcrafted helper functions or post-generation filtering. Additionally, we observe that as the environment designer rapidly increases task difficulty, policy improvement can lag behind, leading to performance plateaus in later iterations. We anticipate that advances in model quality will help mitigate these issues. Future work could focus on improving feasibility – ensuring generated levels are neither too hard nor too easy, for example, by incorporating a minimax regret (Dennis et al., 2020) – and enhancing diversity checks during level generation.

## REPRODUCIBILITY

The Appendix provides further details on our method and experiments. Algorithmic specifics are in Appendix A; simulator implementation details (MiniGrid, PyGame, CARLA) are in Appendix B; LLM prompt examples are in Appendix C; and additional generalization and ablation results are in Appendix D. Our code will be released upon acceptance, while videos and code examples are available as supplementary material.

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

# APPENDIX

The appendix is organized as follows. Section A details the co-evolution algorithm and Nash distribution computation. Section B provides environment implementation specifics for the used simulators (i.e., MiniGrid, PyGame, and CARLA). Section C lists the exact prompts used for environment and policy generation. Section D reports additional experimental results, including generalization and reinforcement learning baselines. Section E presents the best-performing policies per domain. Finally, Section F illustrates examples of evolved environments and their mutation progress.

## A   ALGORITHMIC DETAILS

**Co-Evolution Loop.**    At each generation $t$, the system performs a mutation-based search to synthesize a new environment $\theta_t$ and a corresponding policy $\pi_t$. Environment mutation generates $K$ candidate environments by perturbing the previous one, $\theta_{t-1}$. Each candidate is evaluated under the Nash distribution $w^{t-1}$ over policies from earlier iterations, and the candidate with the lowest expected return is selected. Policy mutation is initialized from the highest-weighted policy $\pi^{\text{best}}$ under $w^{t-1}$. The policy LLM generates $K$ mutated versions of this base policy, which are evaluated solely on $\theta_t$, and the best-performing policy is selected. The pair $(\theta_t, \pi_t)$ is added to the archive, and the payoff matrix $M \in [0,1]^{t \times t}$ is updated accordingly.

**Nash Distribution Computation.**    To determine the current policy mixture, we solve a two-player zero-sum game defined by the empirical payoff matrix $M$. We compute the Nash equilibrium policy distribution by solving the dual linear program of a two-player zero-sum game over the empirical payoff matrix $M$. The optimization is formulated using `PuLP` (Mitchell et al., 2011) and solved using the `CBC` backend solver (Forrest and Ralphs, 2005). The solution yields a probability distribution over policies that minimizes the worst-case environment return:

$$w^t = \arg \max_{w \in \Delta^t} \min_i \sum_j w_j M_{ij}.$$

## B   ENVIRONMENT DETAILS

All environments are being provided, at generation 0, with heuristics to ensure solvability for the specific task. For the environment to be generated, at least one solution is required.

### B.1   MINIGRID IMPLEMENTATION DETAILS

The MiniGrid environment represents a 2D grid-world where each cell encodes the presence of objects, walls, keys, doors, and other entities. The environment supports flexible configurations of size, object placement, and symbolic dependencies, making it suitable for general planning tasks.

**Action Space.**    The agent interacts with the environment using a discrete action space of six primitive actions:

- `TURN_LEFT (0)`: Rotate the agent 90° counterclockwise.
- `TURN_RIGHT (1)`: Rotate the agent 90° clockwise.
- `MOVE_FORWARD (2)`: Advance one tile forward, if the path is free.
- `PICK_UP (3)`: Pick up an object in front, used for collecting keys.
- `DROP (4)`: Drop the currently carried object onto the tile in front.
- `TOGGLE (5)`: Interact with doors in front of the agent:
    - Open a closed door (STATE = 1).
    - Unlock a locked door (STATE = 2) if carrying the correct key.

**Tile Encoding.** Each grid tile is encoded as a 3-tuple of integers:

$$(\texttt{OBJECT\_IDX, COLOR\_IDX, STATE})$$

This structured representation is provided in a fully observable grid array. The indexing is spatial, with `(x, y)` referring to grid row and column, respectively.

Table 2: MiniGrid `OBJECT_IDX` Mappings

| | |
|---|---|
| 0 | Unseen |
| 1 | Empty |
| 2 | Wall |
| 3 | Floor |
| 4 | Door |
| 5 | Key |
| 6 | Ball |
| 7 | Box |
| 8 | Goal |
| 9 | Lava |
| 10 | Agent |

Table 3: Door State Field

| | |
|---|---|
| 0 | Open (passable) |
| 1 | Closed (toggle to open) |
| 2 | Locked (requires key to unlock and toggle) |

**Environment Logic.** Doors and keys are linked by color indices, with up to six distinct colors available. Locked doors block the agent's path until the corresponding key is acquired. The environment enforces procedural placement constraints, ensuring at least one feasible path exists through BFS-based solvability checks. Walls and other obstacles further complicate navigation. The agent maintains a single-key capacity, necessitating key management and path re-planning in multi-door configurations.

**Observations.** At each timestep, the agent receives a fully observable grid state represented as a flattened tensor of shape (`grid_size` $\times$ `grid_size` $\times$ 3), normalized to $[0, 1]$. Each tile encodes the object type, color index, and dynamic state (e.g., door status) as defined by the environment's tile encoding scheme. In addition, the policy receives the agent's absolute position (`agent_pos`) and current orientation (`agent_dir`), enabling precise spatial reasoning and orientation-dependent actions. This structured input enables policies to perform symbolic reasoning without perceptual ambiguity, allowing them to focus solely on decision-making and planning.

### B.2 PyGame Implementation Details

In the PyGame environment, each instance defines a bounded 2D plane in pixel space, with task-specific width and height parameters. The agent is modeled as a circular body with a fixed physical radius of 15 pixels, while the goal zone is a rectangular target area guaranteed to fully contain the agent's circle upon successful completion.

Obstacles are axis-aligned rectangles with randomly positioned and sized dimensions. Their placement follows strict feasibility constraints:

- Obstacles must not overlap with the goal zone.
- Obstacles must not overlap with each other.
- New obstacles are placed only if their inflated bounding box (expanded by the agent's radius) does not intersect existing obstacles, ensuring local non-overlap and feasible placement.

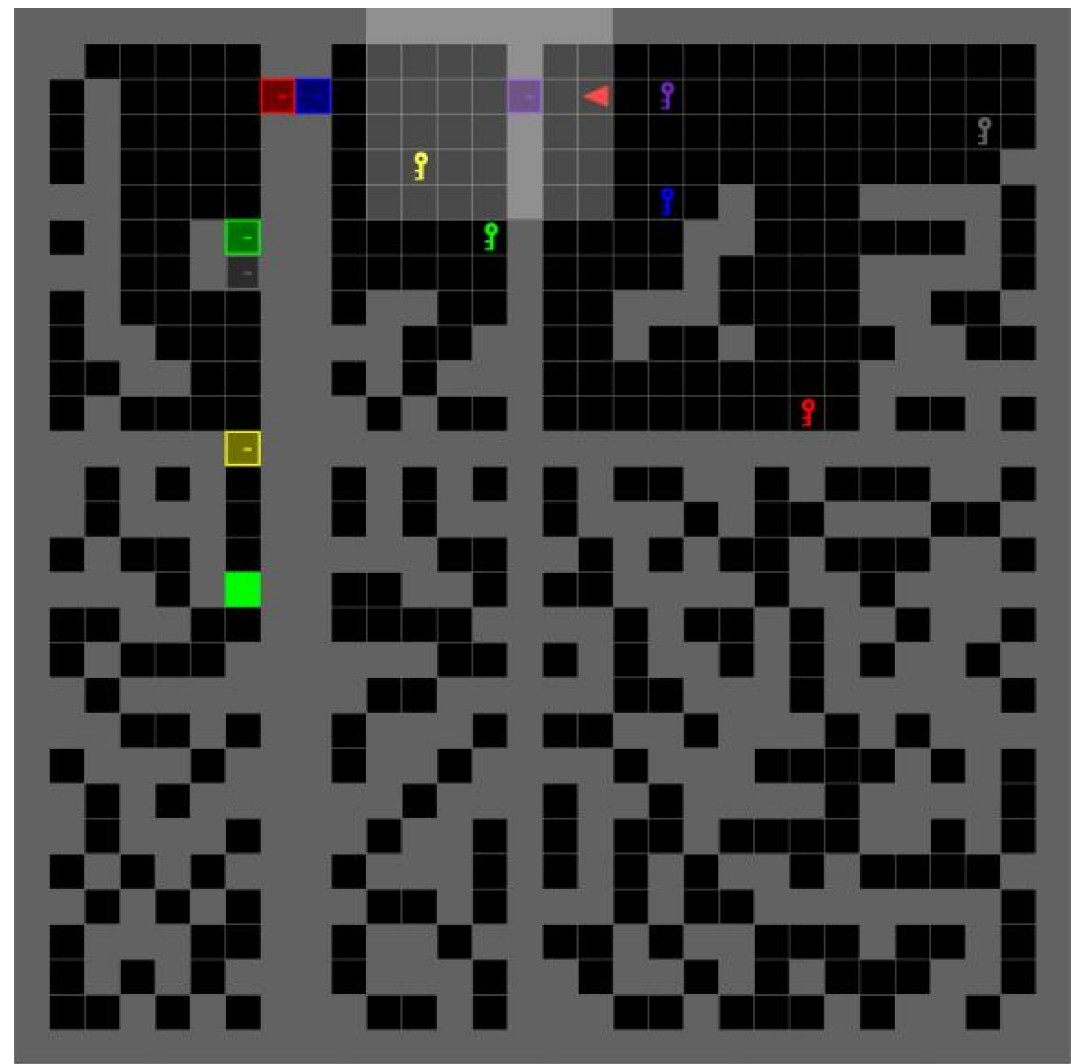

Figure 8: Example of a generated MiniGrid environment (cf. Fig. 3). For this environment, the agent (red arrow) must reach the green goal tile by unlocking the intermediate colored doors using the corresponding keys.

**Action and Observation Spaces.** The agent selects a continuous 2D velocity vector [dx, dy] ∈ $[-1.0, 1.0]^2$ at each timestep. This vector is scaled by an environment-defined speed factor to determine pixel-wise displacement. Collision detection is performed for each proposed movement; invalid moves that would result in obstacle penetration or leaving environment bounds are rejected, leaving the agent stationary.

Observations are provided as a structured dictionary containing:

- `agent_pos`: The agent's center coordinates in pixels.
- `objects`: A list describing the goal zone and each obstacle, with entries specifying `type`, `pos`, `size`, and (for the goal zone) `purpose`.
- `step_count`: The current timestep within the episode.

**Task Parameters.** Task difficulty is progressively scaled by modifying environment parameters, including:

- The number of obstacles, increasing clutter, and requiring more deliberate path planning.

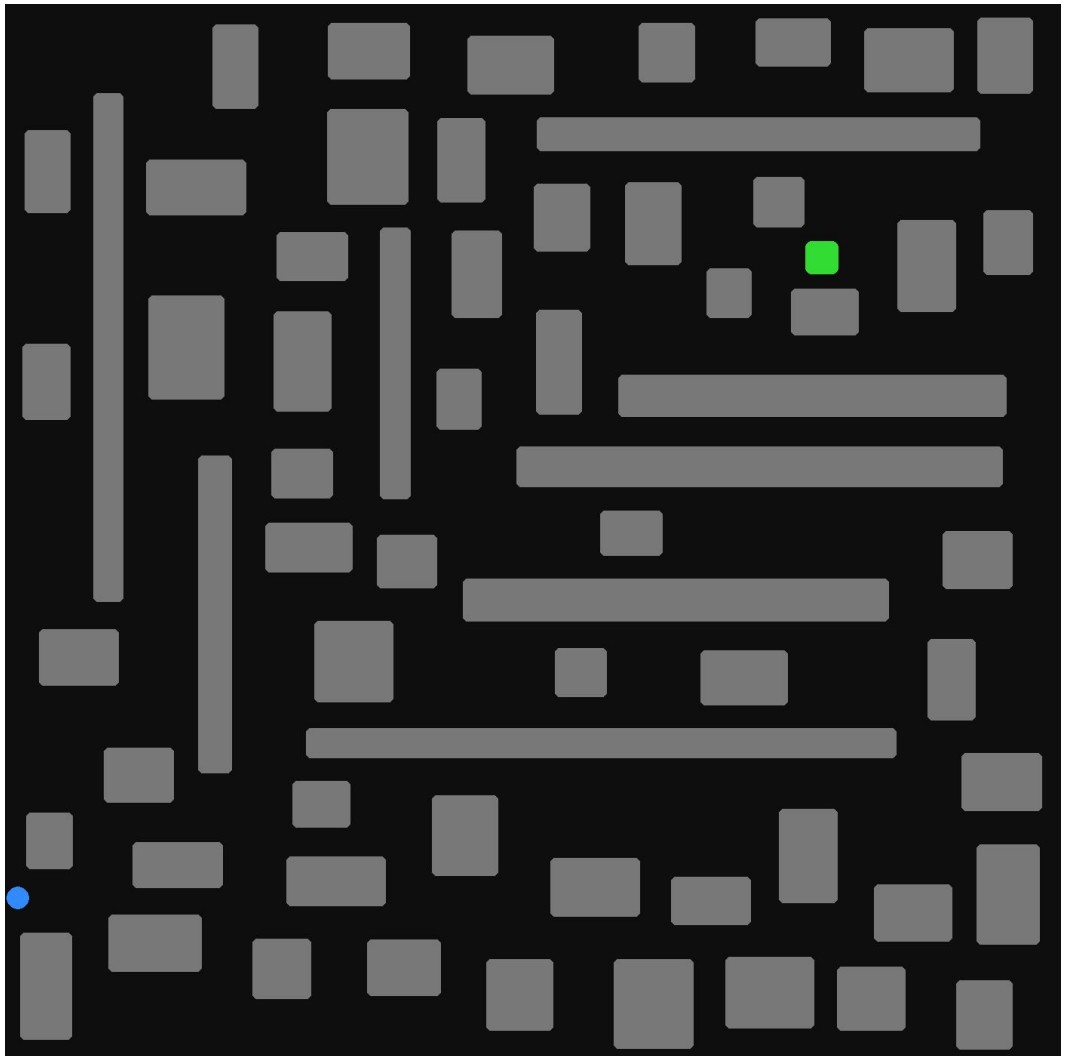

Figure 9: Example of a generated PyGame environment (cf. Fig. 4). In this environment, the agent (blue circle) must navigate the environment spatially to reach the goal (green rectangle).

- The environment's width and height, expanding navigation complexity.
- The agent's movement speed, reducing maneuverability.
- The minimum agent-goal start distance, forcing longer traversal paths.

These parameters are dynamically adjusted by the environment generator to produce increasingly challenging, yet solvable, task instances.

**Episode Termination and Reward.**  An episode terminates when the agent's circular body is entirely within the goal zone or when the maximum allowed steps are exhausted.

**Feasibility Guarantees.**  To ensure the agent can navigate to the goal, the environment performs a reachability check using a discretized occupancy grid that inflates obstacle regions by the agent's radius. This guarantees that all generated tasks are physically feasible for the agent to complete. Invalid placements of obstacles or agent start positions are rejected during the generation process. This process ensures that every evaluation involves meaningful, solvable navigation challenges with non-trivial spatial reasoning requirements.

## B.3 CARLA IMPLEMENTATION DETAILS

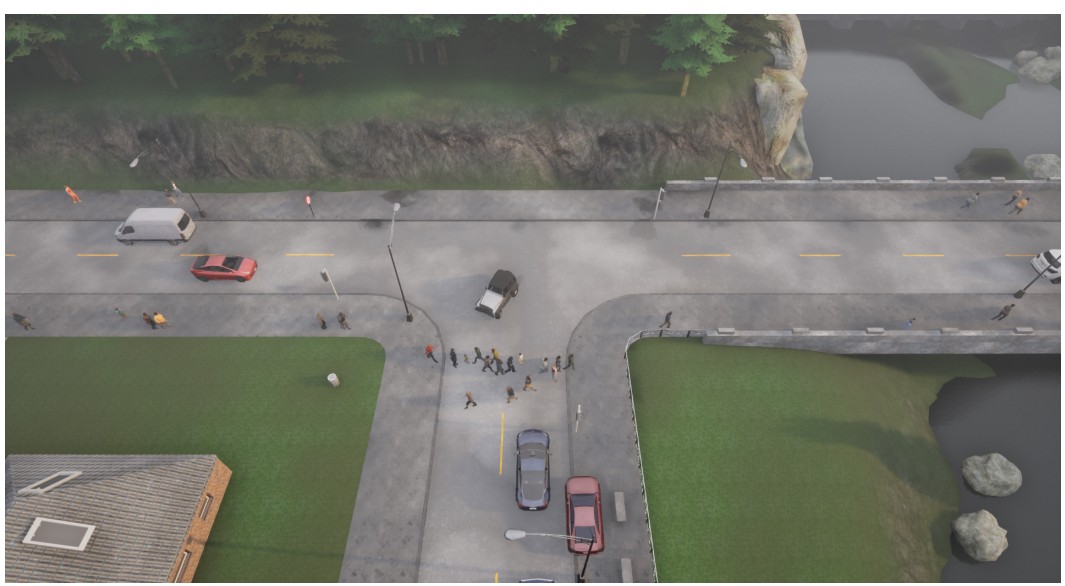

Figure 10: Example of a generated CARLA environment (cf. Fig. 5). From the ego viewpoint, the car perceives pedestrians, traffic lights, and other vehicles as described in Appendix B.3. The red car to the right of the ego vehicle (Tesla v3) was intentionally placed (from the LLM) to confuse the policy.

**Simulator and Map.** We use CARLA `Town01` in synchronous mode with a fixed time step. The route is a recorded closed polyline. The roadway is a two-way single carriageway: one lane per direction, each $\approx 4\,\mathrm{m}$ wide (total $\approx 8\,\mathrm{m}$). Each episode spawns the ego at a fixed start; non-ego vehicles and pedestrians are randomized. Episodes terminate on collision, red-light violation, timeout, or loop completion.

**Frenet Geometry and Progress.** Let the route be a looped polyline $\{P_i\}_{i=1}^N$. For a world point $p \in \mathbb{R}^2$, we project onto each segment

$$t_i = \mathrm{clip}\left( \frac{(p - P_i)^\top (P_{i+1} - P_i)}{\|P_{i+1} - P_i\|_2^2},\ 0,\ 1 \right), \quad \hat{p}_i = P_i + t_i(P_{i+1} - P_i).$$

Let $k = \arg\min_i \|p - \hat{p}_i\|_2$, segment length $\ell_k = \|P_{k+1} - P_k\|_2$, and cumulative arclength up to segment $k$ be $s_k$. We define arclength and lateral offset:

$$s(p) = s_k + t_k\,\ell_k, \qquad \ell_\perp(p) = (p - \hat{p}_k)^\top \mathbf{n}_k, \quad \mathbf{n}_k = \frac{1}{\ell_k}\begin{bmatrix} -d_{k,y} \\ d_{k,x} \end{bmatrix}.$$

Progress from the episode start $s_0$ wraps on the loop: $\Delta s = (s(p) - s_0) \bmod L$, with loop length $L$. We express relative positions/velocities in the ego frame via $R_{\mathrm{we}}(\psi) = \begin{bmatrix} \cos\psi & \sin\psi \\ -\sin\psi & \cos\psi \end{bmatrix}$. The yaw error is $\Delta\psi = ((\psi - \psi_{\mathrm{path}} + \pi) \bmod 2\pi) - \pi$.

**Observation Space.** We expose only the features the policy needs to drive on a prescribed path while interacting with traffic and pedestrians:

*Ego kinematics.* Speed `speed_mps`, yaw rate `yaw_rate_rps`, lateral error $\ell_\perp$, yaw error $\Delta\psi$. Short histories (length 4) for {speed, lateral error, yaw error, past steer/throttle/brake} stabilize control.

*Traffic light.* Nearest traffic light ahead on the route: `exists`, `dist_m`, `state` $\in$ {Red, Green, Yellow}. For simplicity, Yellow is treated as Red.

*Vehicles.* We keep a small, ordered snapshot (top-2) per class with ego-frame gaps and simple surrogates:

$$\text{THW} = \frac{g_x}{\max(0.5, v_{\text{ego}})}, \qquad \text{TTC} = \begin{cases} g_x/(-\Delta v_x), & \Delta v_x < 0 \\ \text{null}, & \text{otherwise.} \end{cases}$$

*Lead cars* are those with Frenet lateral $|\ell_\perp| \leq 2\,\text{m}$ (ego lane). *Opposite cars* fall in $-6 \leq \ell_\perp < -2\,\text{m}$ (oncoming lane).

*Pedestrians.* Within a forward window along the route, we classify: (i) *in-lane* if $|g_y| \leq 2\,\text{m}$; (ii) *approaching* if $2 < |g_y| \leq 3\,\text{m}$ and moving toward the lane ($\Delta v_y g_y < 0$). For approaching walkers, we estimate time-to-enter the near lane edge, $t_{\text{enter}} = (y_\star - g_y)/\Delta v_y$ when $|\Delta v_y|$ is non-negligible, where $y_\star = \pm 2\,\text{m}$.

**Action Space.** A continuous 3-vector (steer, throttle, brake) with steer $\in [-1, 1]$, throttle $\in [0, 1]$, brake $\in [0, 1]$.

**Notes.** This design yields a small, interpretable state while covering path tracking (via $\ell_\perp, \Delta\psi$), car-following and oncoming interactions (via THW/TTC and lane bands), signal compliance (traffic-light snapshot), and pedestrian crossing risk (in-lane vs. approaching with $t_{\text{enter}}$). All constants and implementation details (e.g., horizons, smoothing) are provided in our code release.

# C  PROMPTS

We provide the exact prompts used for environment and policy generation in each domain. These are instantiated dynamically at each iteration, reflecting task-specific parameters and environment configurations.

## C.1  ENVIRONMENT GENERATION PROMPTS

---

### Box 1: MiniGrid Environment Prompt

```
GOAL
Minimize the scalar "Actual Score" in [0,1] evaluated on the Nash-weighted policy mix:
{Weights}
{Policies}

You will return a SINGLE Python class that replaces the existing:
class CustomEnv(MiniGridEnv):

REQUIREMENTS (MANDATORY)
1) Class:
   - Keep "class CustomEnv(MiniGridEnv):" and its public API exactly.
   - Do not modify the base class or inheritance.

2) - If you add helpers, define them inside the same file.
   - Do not rely on undefined globals or external dependencies.

3) Fixed knobs in __init__
   - Set once (and only here):
     self.size = {Size}
     self.num_doors = min({NumDoors}, 6)
   - "_gen_grid()" must use these fixed values directly (no dynamic rescaling).

4) Structured placement
   - Place perimeter, rooms/corridors, doors, keys, and goal via explicit logic.
   - Do not place objects blindly or randomly.

5) Solvability check
   - After placements, call "check_solvable(self, start, goal)" exactly as provided.
   - Accept the layout only if it returns True.

6) Episode diversity
   - "_gen_grid()" must generate different layouts across episodes (vary partitions, door indices,
     key pockets) while using the fixed knobs.

7) Termination
   - Episode ends when (a) the agent reaches the goal, or (b) max_steps is exceeded.

8) Retry policy
   - If unsolvable, retry up to 1000 times.
   - If still unsolvable, set "self.failure_feedback" and return.

FEASIBILITY FUNCTIONS (DO NOT MODIFY)
- "_verify_mandatory_door_keys(self)"
- "bfs_ignore_doors(self, start, goal)"
- "bfs_block_locked(self, start, goal)"
- "_find_key_spot_block_locked(self, agent_pos, door_pos, unlocked_colors)"
Use these as-is. You may add helpers, but never replace or alter them.

OUTPUT
Return ONLY the updated "CustomEnv" class (no commentary).
```

---

```
Box 2: PyGame Environment Prompt

GOAL
Minimize the scalar "Actual Score" in [0,1] evaluated on the Nash-weighted policy mix:
{Weights}
{Policies}

REQUIREMENTS (MANDATORY)
1) Class/API
   - Keep "class CustomEnv:" and its public API exactly.
   - Do not change the class name or inheritance.

2) Implement only these methods:
   - reset(self)
   - step(self, action)
   - draw_objects(self)
   - task_description(self)
   - _get_obs(self)
   - render(self)
   - Any private helpers defined inside the class (e.g., _handle_quit, _sample_pos).

3) task_description(self)
   - Must return a plain string describing:
     - The task objective (agent must reach the goal zone).
     - The action space (continuous [dx,dy] in [-1.0,1.0]^2).
     - Key parameters (sizes, margins, speeds).
     - The full observation dictionary structure.

4) Episode termination
   - Ends if agent reaches the goal (checked externally by _check_done()).
   - Ends if max_steps is exceeded.
   - Do not call _check_done() inside step().

5) Structured placement
   - Place agent, goal, and obstacles via explicit rules.
   - Ensure no overlaps; keep all objects inside bounds.
   - Guarantee solvability (always at least one valid path).

6) Randomization
   - Use structured randomness (np.random.randint, np.random.uniform) in reset().
   - Every reset() must produce a distinct environment instance.
   - Randomness must contribute to meaningful diversity.

7) Safety
   - The goal zone must be large enough: width,height >= 2*agent_radius + margin.

8) Behavior
   - step(action): interpret action as 2D continuous move.
   - Call self._handle_quit() at the start to process quit events.
   - Return (obs, reward, done) where obs=self._get_obs(), reward=0.0, done=self.done.

9) Observations
   - _get_obs() must return:
     {
       "agent_pos": [x,y],
       "agent_radius": r,
       "objects": [
         { "type":"zone", "pos":[cx,cy], "size":[w,h], "purpose":"goal" },
         { "type":"obstacle", "shape":"rect", "pos":[cx,cy], "size":[w,h] },
         ...
       ],
       "bounds": [W,H],
       "step_count": N,
       "max_steps": M
     }

10) Rendering
   - draw_objects(): use pygame primitives; only draw if self.render_mode==True.
   - render(): create/update PyGame surface, call draw_objects(), flip buffers if render_mode==True.

TASK EVOLUTION
   - Increase distance between agent and goal.
   - Add obstacles or tighter passages.
   - Increase W,H and proportionally increase max_steps.
- You are free and motivated to introduce new difficulties as long as the task remains the same: The
     agent must reach the goal zone.

CONSTRAINTS
- Do NOT add symbolic puzzles (no keys, doors, colors).
- Do NOT use MiniGrid tile logic.
- Do NOT add irrelevant randomness.
- Do NOT remove or rename required methods.
- Do NOT alter the external _check_done().

OUTPUT
Return ONLY the updated "CustomEnv" class (no commentary).
```

## Box 3: CARLA Environment Prompt

```
GOAL
Minimize the scalar "Actual Score" in [0,1] evaluated on the Nash-weighted policy mix:
{Weights}
{Policies}
Current environment performance: {ActualScore}

You will return a SINGLE Python class that replaces the existing:
{Actual_Class}

REQUIREMENTS (MANDATORY)
1) Class
   - Keep the class name "CarlaTown01Env" exactly.
   - Preserve all existing methods; you may add new helpers inside the class.

2) Task identity
   - Ego vehicle is always vehicle.tesla.model3.
   - Start and goal follow the same Town01 loop.
   - Do not spawn oversized vehicles (buses, trucks) or actors that may block solvability.

3) task_description(self)
   - Must return a plain string describing:
      - The driving objective (complete the loop without collisions).
      - Key environment elements (traffic, pedestrians, lights, dynamic behaviors).
      - The observation dictionary fields provided to the policy.
   - If new fields are added to obs (via get_obs), they must be explicitly documented in this string.
   - Do not remove fields; only extend if needed.

4) Observations
   - get_obs() must remain consistent with the description.
   - The observation dictionary includes ego state, histories, traffic lights, lead cars, opposite
     cars, pedestrians.
   - New factors (e.g., jaywalkers, lane changes) must be added carefully and described in
     task_description.

5) Solvability and safety
   - Always ensure at least one feasible driving strategy exists.
   - Pedestrian side-hit guard must remain intact.
   - Adjust max_steps proportionally if difficulty increases.
   - No actor may force unsolvable collisions.

6) Constraints
   - Do not add global code or side effects outside the class.
   - Do not remove feasibility checks already in place.
   - Do not change the class name.

OUTPUT
Return ONLY the updated CarlaTown01Env class (no commentary).
```

## C.2 POLICY GENERATION PROMPTS

---

### Box 4: MiniGrid Policy Prompt

```
GOAL
You are tasked with improving an existing policy function for navigating MiniGrid environments.
The policy must analyze the grid, reason about objects, plan an optimal path, and execute actions
    efficiently.
The objective is to reach the goal tile (OBJECT_IDX=8).

You are provided with:
- Actual Score = {ActualScore}, a scalar in [0,1] that reflects the performance of the given policy.
- Policy = {Policy}, the current implementation of the policy function.

YOUR TASK
Analyze the given policy together with its score and modify it to improve performance.
The output must be a new version of the same function with improvements.

OUTPUT
Return a single Python function:
def policy(obs, agent_pos, agent_dir):  # -> int in {0,1,2,3,4,5}

ENVIRONMENT FORMAT
- obs is a 2D NumPy array of shape (grid_size, grid_size, 3).
- Each tile is encoded as (OBJECT_IDX, COLOR_IDX, STATE).
- Indexing is (x=row, y=column).

OBJECT_IDX MAP: 0=Unseen, 1=Empty, 2=Wall, 3=Floor, 4=Door, 5=Key, 6=Ball, 7=Box, 8=Goal, 9=Lava, 10=
    Agent

DOOR STATE: 0=Open (free to pass), 1=Closed (requires Toggle action=5 when facing), 2=Locked (
    requires correct key + Toggle=5)

ACTIONS: 0=Turn Left, 1=Turn Right, 2=Move Forward, 3=Pick Up, 4=Drop, 5=Toggle

STRICT REQUIREMENTS
1) Goal-Oriented Navigation
- Always plan and execute a valid path to the Goal (OBJECT_IDX=8).
- Avoid unnecessary detours unless a locked door blocks the path.

2) Door Handling
- Open doors (STATE=0) act as free space.
- Closed doors (STATE=1): face the door, Toggle (5) to open, then Move Forward (2).
- Locked doors (STATE=2): only approach after collecting the correct key. Face the door, Toggle (5),
    then Move Forward (2).

3) Key Handling
- Keys are only collected if required to unlock a blocking door.
- The agent can hold exactly one key at a time.
- If already holding a different key, Drop (4) into the front cell (if empty) before picking up the
    new one.
- Keys must be picked up with Pick Up (3) when the agent is adjacent and facing the key.
- Dropped keys must remain accessible.

4) Safety and Obstacles
- Never Move Forward (2) into a Wall (2) or Lava (9).
- Treat Unseen tiles (0) as blocked until explored.

5) Orientation
- Before any interaction (Move Forward, Pick Up, Drop, Toggle), ensure the agent is facing the
    correct adjacent cell.
- Rotate (0=Left, 1=Right) until aligned, then act.

6) Termination
- The episode ends when the agent reaches the Goal or exceeds max_steps.
- The policy must minimize wasted actions and maximize efficiency.

EDGE CASES
- If the agent needs a key but already holds another, drop the held key before pickup.

FORBIDDEN
- Changing the function name, arguments, or return type.
- Returning values outside {0,1,2,3,4,5}.
- Using randomness, global state, or external libraries.

OUTPUT FORMAT
- Return only the improved function `def policy(...)` in valid Python.
- No explanations, no comments, no extra text.
```

**Box 5: PyGame Policy Prompt**

```
GOAL
Maximize the scalar "Actual Score" in [0,1] for the current policy by improving the given function.

YOU MUST RETURN
A SINGLE Python function that replaces the existing one:
def policy(obs):  # -> [dx, dy] in [-1.0, 1.0]^2

INPUTS
- Actual Score = {ActualScore}
- Given Policy = {Policy}
- Observation dictionary schema (exact field names and meaning) = {obs_dict}

FUNCTION CONTRACT
- Keep the exact signature: def policy(obs).
- Return a 2D continuous action [dx, dy] with each component in [-1.0, 1.0].
- Do not use randomness, globals, I/O, or external libs beyond numpy.

OBSERVATION DICTIONARY
- Use only fields provided in obs. The runner supplies {obs_dict}. The typical structure is:
  - agent_pos: [x, y]  current agent center position in pixels
  - agent_radius: r  agent circle radius in pixels
  - objects: list of dicts describing scene items. Each item:
    - type: "zone" or "obstacle"
    - pos: [cx, cy]  center position
    - size: [w, h]  rectangle width, height
    - purpose: optional string, e.g., "goal" for the goal zone
  - bounds: [W, H]  canvas width and height in pixels
  - step_count: N  steps elapsed
  - max_steps: M  step limit before timeout

STRICT REQUIREMENTS
1) Goal-directed motion
   - Move toward the goal zone whenever a safe direction exists.
   - Prefer progress that reduces distance-to-goal.

2) Obstacle and bounds safety
   - Do not output actions that would cause collision with any obstacle rectangle.
   - Do not move outside [0, W] x [0, H].
   - Respect agent_radius clearance when deciding direction.

3) Action validity
   - Clamp or normalize output so each component stays within [-1.0, 1.0].
   - Avoid jitter or oscillation near the goal.

4) Robustness
   - If the direct route is blocked, choose a safe detour around obstacles.
   - Avoid loops by preferring actions that reduce distance-to-goal over time.

EDGE CASES
- Goal directly reachable: head straight toward the goal center or safe entry edge.
- Narrow passage: align with passage axis and pass through without scraping boundaries.
- Stuck against obstacle: choose an alternate heading that increases free-space margin.
- Near goal zone edge: reduce overshoot and enter the zone cleanly.

QUALITY TARGETS
- Short time-to-goal.
- Minimal wasted motion and reversals.
- Collision-free trajectories across diverse layouts.

FORBIDDEN
- Changing the function name, arguments, or return type.
- Returning values outside [-1.0, 1.0].
- Ignoring obstacles, bounds, or agent_radius in decisions.
- Using randomness, global state, file or network I/O, or non-numpy libraries.

OUTPUT
Return ONLY the improved function "def policy(obs):"
No explanations, no comments, no extra text.
```

## Box 6: CARLA Policy Prompt

```
GOAL
Maximize the scalar "Actual Score" in [0,1] by improving the current driving policy.

You will return a SINGLE Python class that replaces the existing:
{Actual_Policy}

INPUTS
- Actual Score = {ActualScore}
- Previous Policy = {Policy}
- Path = np.ndarray (N,2) lane-center polyline

STRICT REQUIREMENTS
1) Class/API
   - Keep the class name "Policy".
   - Implement __init__(self) and compute_action(self, obs, path).
   - Return (steering, throttle, brake) as floats.
   - steering in [-1,1], throttle in [0,1], brake in [0,1].
   - If brake > 0 then throttle must equal 0.

2) Determinism and smoothness
   - No randomness or learning.
   - Ensure gradual changes, avoid jerks.

3) Robustness
   - Handle None or NaN conservatively.
   - On invalid input, default to safe stop (steer=0, throttle=0, brake>0).
   - No prints or logging.

4) Sign conventions
   - lateral_hist4: right-positive meters.
   - yaw_error_hist4: ego yaw - path yaw, right-positive.
   - yaw_rate_rps: right-positive radians/s.

OBSERVATION FORMAT
obs is a dictionary containing:

- Ego state:
  - speed_mps: current speed in m/s
  - yaw_rate_rps: yaw rate in rad/s (right-positive)

- Ego histories (arrays of length 4):
  - speed_hist4: past speeds
  - lateral_hist4: lateral errors (m, right-positive)
  - yaw_error_hist4: yaw errors (rad, right-positive)
  - steer_cmd_hist4: previous steering commands
  - throttle_cmd_hist4: previous throttle commands
  - brake_cmd_hist4: previous brake commands

- Traffic light:
  - exists: boolean
  - state: int {0=unknown, 1=green, 2=yellow, 3=red}
  - dist_m: distance to stop line (m)

- Lead cars (up to 2, same schema each):
  - gap_long_m: longitudinal gap (m)
  - gap_lat_m: lateral gap (m)
  - rel_long_mps: relative longitudinal speed (m/s)
  - ttc_s: time-to-collision (s)
  - thw_s: time headway (s)

- Opposite cars (up to 2, same schema as lead cars)

- Pedestrians (variable count):
  - lane: lane index
  - state: int encoding motion state
  - gap_long_m: longitudinal gap (m)
  - gap_lat_m: lateral gap (m)
  - rel_lat_mps: relative lateral speed (m/s)
  - t_enter_lane_s: predicted time to enter lane (s)
  - side: which side of road (left/right)

- Perception horizon:
  - All dynamic actors truncated to 35 m ahead of ego

OBJECTIVES
- Lateral: minimize lateral and yaw errors relative to the centerline.
- Longitudinal: track target speed up to 6.94 m/s (25 km/h) if unimpeded.
- Traffic lights: stop smoothly before stop line on red; never cross on red.
- Pedestrians: yield to pedestrians in or entering ego lane.
- Lead vehicles: maintain safe following distance; avoid indefinite blocking.
- Precedence order: red light stop > pedestrian yielding > lead vehicle following > cruising.
- Fail-safe: if uncertain, perform controlled stop.
- Comfort: avoid abrupt oscillations; prioritize smooth steering and braking.

FORBIDDEN
- Changing the class name or method signatures.
- Returning values outside steering/throttle/brake ranges.
- Simultaneous throttle and brake > 0.
- Using randomness, logging, prints, or external dependencies.

OUTPUT
Return ONLY the improved class "Policy".
No explanations, no comments, no extra text.
```

# D  ADDITIONAL RESULTS

## D.1  GENERALIZATION ACROSS ENVIRONMENTS

We compare *standardized, unseen environments* to those produced during co-evolution. The standardized set comprises MiniGrid `DoorKey-16x16-v0` and `LockedRoom-v0`, and CARLA `Town02` (trained on `Town01`). These differ from our evolved environments in three respects: (i) *structure* (fixed layouts and goal semantics rather than co-evolved variants), (ii) *scale* (grid/world size and path lengths), and (iii) *sequential dependencies* (e.g., key–door ordering and room unlocking). For CARLA, `Town02` diverges from `Town01` by road-network density and traffic complexity: it has sharper turns, narrower lanes, and more intersections and pedestrian crossings, requiring longer detours and tighter maneuvers compared to the more regular Town01 layout. We evaluate with identical rollout settings.

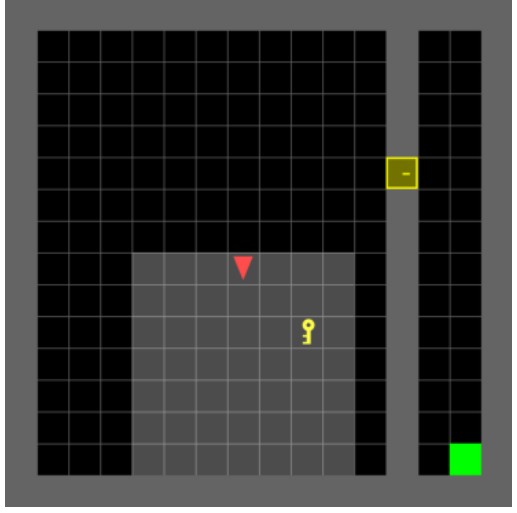

(a) `DoorKey-16x16-v0` (MiniGrid)

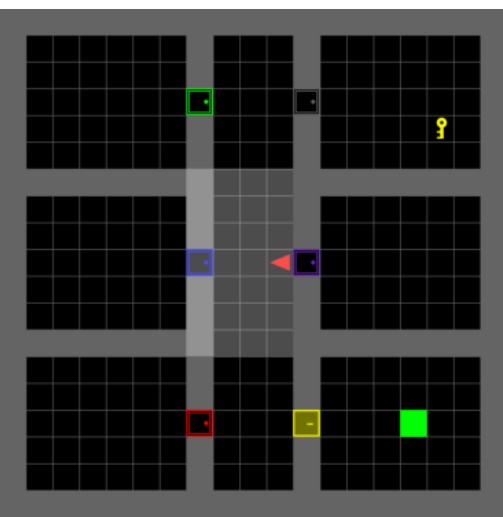

(b) `LockedRoom-v0` (MiniGrid)

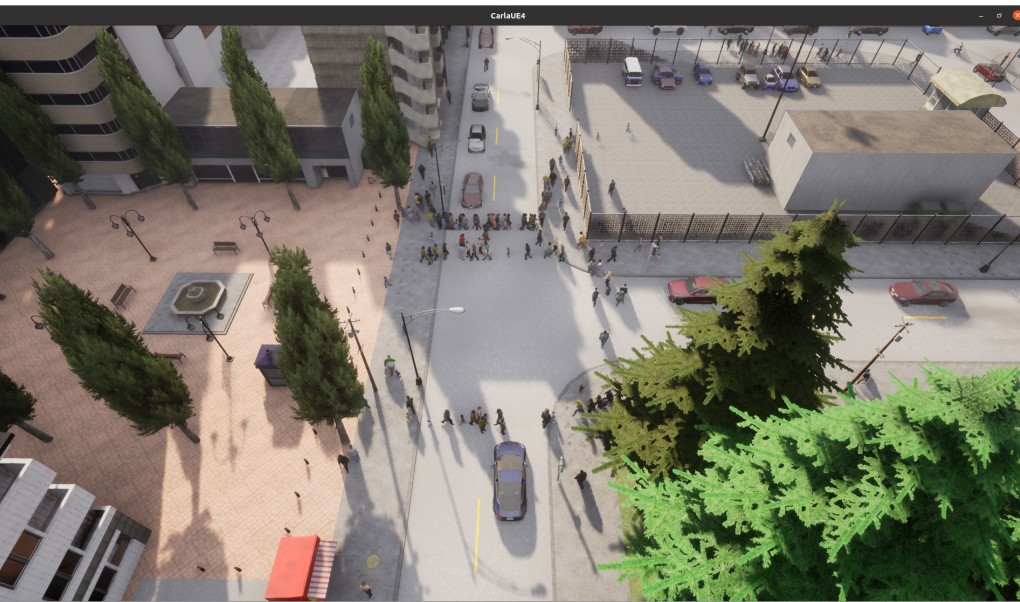

(c) `Town02` (CARLA)

Figure 11: Examples of previously unseen standardized environments used to validate generalization. MiniGrid snapshots (*top*) and CARLA `Town02` (*bottom*).

## D.2 Reinforcement Learning Results

### D.2.1 MiniGrid Maze-solving

We evaluate two representative algorithms using Stable-Baselines3 Raffin et al. (2021a): PPO, a policy-gradient method, and QRDQN, a value-based method for discrete domains.

**Reward shaping.** For reward, we use the default MiniGrid reward function:

$$R(s,a) = \begin{cases} 1 - 0.9 \cdot \dfrac{t}{T_{\max}}, & \text{if the agent reaches the goal at step } t, \\ 0, & \text{otherwise,} \end{cases}$$

where $T_{\max}$ is the maximum episode length. Thus, faster completion yields a higher return.

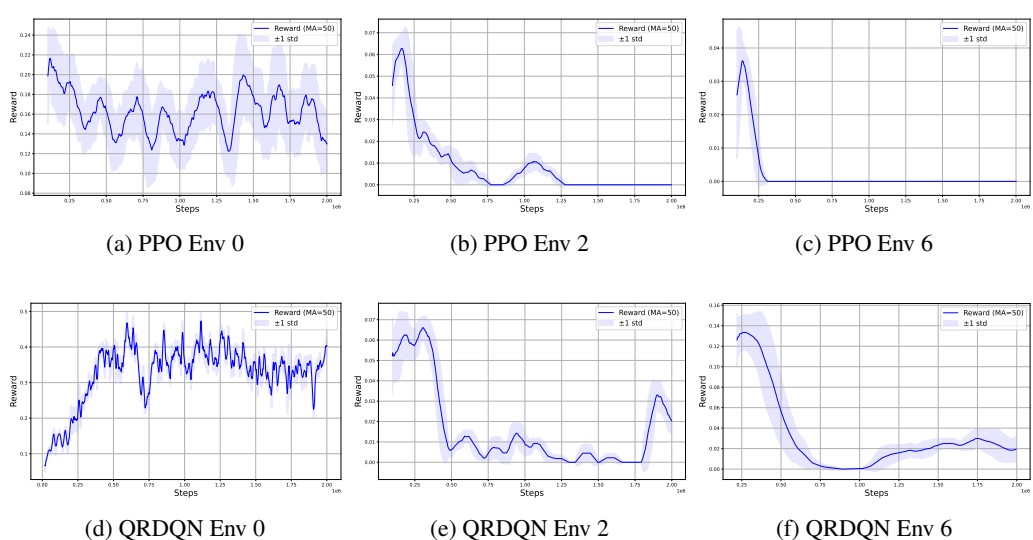

(a) PPO Env 0      (b) PPO Env 2      (c) PPO Env 6

(d) QRDQN Env 0      (e) QRDQN Env 2      (f) QRDQN Env 6

Figure 12: Training curves of PPO (*top*) and QRDQN (*bottom*) across MiniGrid environments. The y-axis represents reward, and the x-axis represents total training steps.

| Algorithm | Env 0 | Env 2 | Env 6 |
|---|---|---|---|
| PPO | $12.0 \pm 1.4\%$ | $0.0 \pm 0.0\%$ | $0.0 \pm 0.0\%$ |
| QRDQN | $68.5 \pm 12.0\%$ | $0.0 \pm 0.0\%$ | $0.0 \pm 0.0\%$ |

Table 4: Success rates (%, mean $\pm$ std over two runs) across MiniGrid environments.

### D.2.2 PyGame 2D Navigation

We evaluate two representative algorithms using Stable-Baselines3 Raffin et al. (2021a): PPO, and SAC.

**Reward shaping.** In the PyGame environments, reward is sparse with a per-step penalty:

$$R(s,a) = \begin{cases} +1, & \text{if the agent reaches the goal,} \\ -0.01, & \text{otherwise (each step).} \end{cases}$$

This encourages agents to minimize path length while ensuring sparse success feedback.

## E Best Performing Policies

The final evolved policies are too extensive to analyze line by line. Instead, we provide high-level summaries of their algorithmic structure and key heuristics.

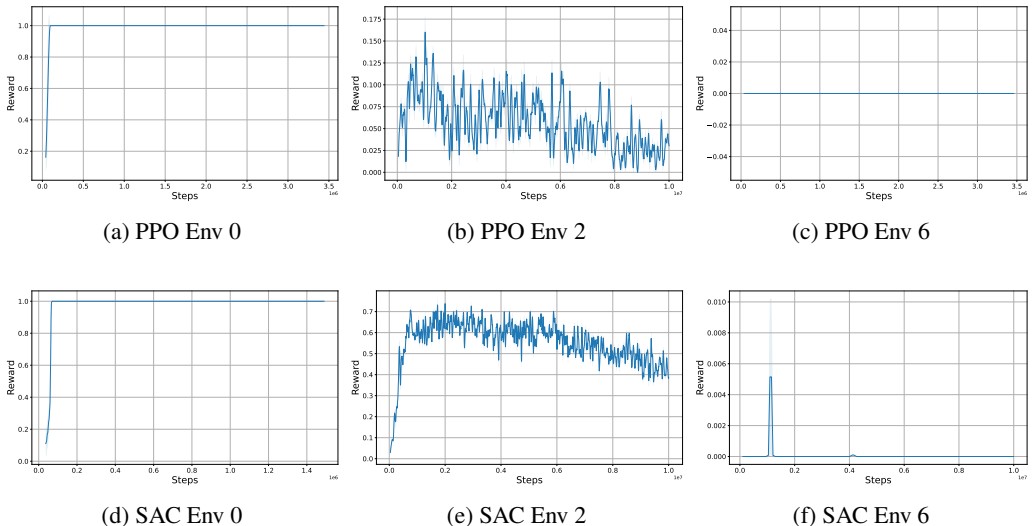

Figure 13: Training curves of PPO (top) and SAC (bottom) across PyGame environments. The y-axis represents reward, and the x-axis represents total training steps.

| Algorithm | Env 0 | Env 2 | Env 6 |
|---|---|---|---|
| PPO | $61.7 \pm 2.3\%$ | $6.2 \pm 0.7\%$ | $0.0 \pm 0.0\%$ |
| SAC | $88.2 \pm 3.0\%$ | $22.5 \pm 5.9\%$ | $0.0 \pm 0.0\%$ |

Table 5: Success rates (%, mean $\pm$ std over two runs) across PyGame environments.

### E.1 BEST PERFORMING MINIGRID POLICY

The best performing policy (`policy_8`, see Figure 6) follows a structured four-stage loop: (1) immediate interaction with the tile in front (e.g., toggling doors, picking up keys), (2) direct *A\* pathfinding* through open doors only, (3) targeted handling of blocking doors by acquiring and using the correct key, and (4) mild exploration when no other action is available.

A key improvement is the **important-cell-aware key dropping** routine, which ensures that keys are never dropped on critical tiles such as the goal, door-adjacent cells, or bottleneck corridors, and includes a cooldown to prevent repeated drops at the same location.

These components result in a reactive decision agent that can reliably plan paths, resolve sequential door–key dependencies, and avoid pathological key-handling loops in complex mazes.

### E.2 PYGAME POLICY

Notable improvements for the best performing policy (`policy_8`, see Figure 6) for *the PyGame 2D navigation* task are as follows:

- **Hybrid global–local navigation:**
  1. Global *A\* pathfinding* on a cached occupancy grid with inflated obstacles, ensuring feasibility given the agent's radius.
  2. *Line-of-sight smoothing* and density-based waypoint pruning to shorten paths and reduce oscillations.
  3. Local *predictive collision checking* before every step, with sidestep maneuvers when obstacles block the planned motion.
  4. Re-planning trigger when pinned or when progress stalls for multiple steps.
- **Occupancy grid acceleration**: Inflated obstacle rectangles are pre-painted into a coarse grid, avoiding repeated per-step collision checks and reducing computational overhead.

- **Stuck-resolution heuristics**: A deterministic sidestep direction is remembered to avoid ping-pong loops, and a lightweight jitter fallback explores when no path can be planned.
- **Precision finishing**: When near the goal, the agent switches to a direct centering vector for reliable termination upon goal overlap.

These improvements yield a *planning-reactive agent* that, for every step: **1)** computes a global path using A* with cached obstacle inflation, **2)** prunes and smooths waypoints, **3)** performs predictive collision checks with local sidesteps, and **4)** re-plans when progress halts. This combination enables robust navigation even in cluttered environments with narrow corridors.

### E.3 CARLA POLICY

The best performing controller (`policy_9`, see Fig. 6) augments a smooth cruise/follow core with a clearance-aware passing routine and stricter intersection handling.

**Four-stage loop.** (1) **Signal gating**: strict traffic-light guard (*Yellow = Red*), stop-line latch, and pedestrian holds; approach speed is limited by both stop-line distance and queued-lead gap. (2) **Lead classification**: distinguishes a right-curb parked blocker from an in-lane stopped lead using lateral intrusion and relative speed cues. (3) **Clearance-aware pass**: if the blocker is parked, oncoming is clear, and distance gates are met, the agent enters a bounded left-offset pass. It maintains a minimum offset and a small, gated opposite-lane incursion, *holds* the offset while alongside, and only recenters after front-clearance (with a brief hold if the lead vanishes). (4) **Smooth tracking**: target-speed smoothing with curvature/heading caps and a damped lookahead lateral controller; unstick logic provides a gentle creep when safe.

**Key heuristics.**

- **Stop-line priority**: combined stop-line/lead-gap caps and a near-line latch prevent creeping over the line on non-green states.
- **Right-edge pass safety**: minimum pass offset, centerline guard, and oncoming no-pass gate; tiny opposite-lane incursion is permitted only when clear.
- **Stability under occlusion**: offset-hold on brief lead dropouts avoids snap-back; post-clear recenter includes a short hysteresis.

## F EVOLVED ENVIRONMENTS

### F.1 MINIGRID ENVIRONMENT MUTATIONS

In the MiniGrid maze-solving task, the LLM mutates discrete grid-worlds where an agent must navigate to a goal while avoiding obstacles, doors, and keys. Difficulty increases through the following mechanisms:

- **Grid scaling**: larger grids extend path length and increase exploration requirements.
- **Obstacle density**: additional walls create more complex mazes and reduce direct visibility of the goal.
- **Sequential key–door dependencies**: locked doors are introduced along the main corridor, requiring keys to be collected and used in the correct order.
- **Hard vs. soft chokepoints**: some doors are reinforced by barrier walls that force strict bottlenecks, while others include short wings or detours that add complexity without fully blocking the corridor.
- **Protected corridors**: a one-cell halo ensures that critical key–door paths remain open even as random obstacles are added, guaranteeing solvability.

This progression transforms initially trivial layouts into structured mazes that demand multi-step reasoning, ordered dependencies, and long-horizon planning while ensuring that every environment remains solvable by construction.

## F.2 PYGAME ENVIRONMENT MUTATIONS

In the PyGame navigation task, the LLM mutates a continuous 2D arena where a circular agent must reach a rectangular goal zone while avoiding collisions. While early generations adjust simple parameters such as arena size or obstacle counts, later environments evolve into structured mazes with corridor-like passages and long detours. Key mutation axes include:

- **Corridor formation**: long rectangular bars are placed to partition the arena into corridors, forcing agents to identify traversable passages rather than rely on direct routes.
- **Bottleneck and detour creation**: increasing bar thickness and obstacle density narrows passageways and introduces dead ends, requiring agents to plan long, non-greedy paths.
- **Start–goal separation**: minimum distance constraints push the agent to begin far from the goal, ensuring navigation requires multiple turns and obstacle avoidance.
- **Precision termination**: the goal region remains small relative to agent size, demanding careful alignment to trigger success.
- **Scalable horizons**: enlarging arenas and increasing maximum steps allows environments to grow in complexity without becoming unsolvable.

Unlike gridworlds, these continuous PyGame arenas induce navigation behaviors closer to geometric planning: agents must balance global pathfinding with local collision checks, and later evolved environments present rich mazes with narrow corridors that mimic real-world navigation challenges.

## F.3 CARLA ENVIRONMENT MUTATIONS

In the *Carla Town01* driving task, the LLM mutates a fixed urban loop with signalized intersections, oncoming traffic, and pedestrians. Difficulty rises from light, compliant flows to dense, heterogeneous traffic with narrow-clearance segments, while remaining solvable by construction.

- **Traffic scaling**: vehicle counts increase from light to heavy urban load; speed variance and lane changes introduce realistic flow heterogeneity.
- **Pedestrian pressure**: higher crossing rates and tighter cadences create frequent curb-to-lane interactions requiring cautious approach and yielding.
- **Intersection strictness**: virtual "second gates" beyond stop lines mirror light states, penalizing early acceleration and forcing disciplined red/yellow behavior.
- **Narrow-clearance segments**: parked or frozen intrusions create lane squeezes that demand bounded lateral offsets and precise, short opposite-lane incursions when clear.
- **Micro-perturbations**: periodic brake-taps on leads and occasional temporary stoppers test following stability without causing deadlocks.
- **Oncoming dynamics**: faster opposite-lane bursts create brief no-pass windows, requiring agents to time passes and maintain centerline guards.
- **Jam watchdog & solvability**: stall detectors inject bounded flow perturbations to unstick traffic; obstacle placements and signal logic are constrained to ensure episodes remain completable.
- **Observation compatibility**: added features (e.g., lane-squeeze indicators, extended stop-line states) are exposed via backward-compatible fields to avoid policy breakage.

This progression turns a benign city loop into a dense, signal-rich scenario with tight margins and bursty interactions, pushing policies to coordinate cautious intersection handling, safe passing, and recovery from transient jams.

