# OpenReview forum: "Adversarial Co-Evolution of LLM-Generated Policies and Environments via Two-Player Zero-Sum Game"
_ICLR.cc/2026/Conference — ICLR 2026 Conference Withdrawn Submission_

### Official Review · Reviewer_6dck · 2025-10-31

**Soundness:** 2
**Presentation:** 3
**Contribution:** 2
**Rating:** 4
**Confidence:** 3

**Summary:**

The paper proposes Covolve, where an LLM generates both environments and policies as executable code. As a two-player zero-sum game, the method computes a mixed-strategy Nash equilibrium (MSNE) over the generated environments and policies, inducing them to co-evolve toward increasing complexity.

**Strengths:**

Using an MSNE helps robust co-evolution across the generated environments, rather than overfitting to the most recent one.

**Weaknesses:**

* As iterations proceed, the number of stored policies and environments grows, making payoff-matrix evaluation and MSNE solving increasingly expensive.
* Because the environment designer aims to increase difficulty, it may generate environments that are too hard to learn or even unsolvable.
* Generating policies and environments with an LLM requires prompt engineering and post-generation filtering.
* Empirical comparisons focus on Eurekaverse; other UED baselines (e.g., [1]) are not compared.

[1] Jiang et al., "Replay-Guided Adversarial Environment Design", 2021

**Questions:**

* How were the prompts in Appendix C designed and selected?
* What fraction of generated environments or policies are filtered?

---

### Official Review · Reviewer_XFc4 · 2025-11-01

**Soundness:** 2
**Presentation:** 3
**Contribution:** 2
**Rating:** 2
**Confidence:** 3

**Summary:**

- The authors propose a method that iteratively improves both the policy and the environment using LLM-based code generation.
- They apply PSRO’s multi-agent framework by modeling the policy and environment as a two-player game.
- The ultimate goal is to produce a policy robust to diverse environments.

**Strengths:**

- To address the forgetting issue in the co-evolution framework, the authors propose storing both the environment set and the policy set.

**Weaknesses:**

- Modeling agent-environment co-evolution as a zero-sum game may not be the most suitable formulation.
    - As noted in the paper, the key goal in environment design is to create environments that are solvable yet appropriately challenging.
    - However, the proposed method, framed as a zero-sum (minimax) game, can generate unsolvable environments.
    - Even with heuristic solvability filtering, the following issues may arise:
        - During iteration, a $\theta_i$ may be sampled that most policies cannot solve.
        - All policies in $P$ yield $U = 0$ for $\theta_i$.
        - The solution to Eq. 1 becomes non-unique, corrupting the MSNE policy.
        - Consequently, meaningful environments and policies become difficult to generate in subsequent iterations.
- Unfair comparison with Eurekaverse.
    - While it is acknowledged that forgetting can occur in Eurekaverse, the original method does not update the agent and environment in a zero-sum game manner. In contrast, this paper implements it using a zero-sum formulation.
    - As discussed in the above weakness, the zero-sum game approach risks destabilizing learning, which likely accounts for the degraded performance of Eurekaverse in the experiments.
    - Therefore, the version of Eurekaverse implemented in this paper differs significantly from the original; it would be more appropriate to report its results under a new name, such as **CoEvolve-Naive**, and it is required to re-conduct comparisons with the original Eurekaverse under equivalent experimental settings.
- Storing policy and environment sets, necessary for mitigating forgetting, incurs increased computational complexity.
    - As iterations progress, evaluation time for policies and environments grows, leading to quadratic overall time complexity.
    - Additionally, deploying the obtained MSNE policy requires storing the entire policy set.
- Figure 2 caption claims the proposed method guarantees solvable environments, but no such contribution exists in the main text.

**Questions:**

- The RL comparison in Section 5.4 (Ablation Study) seems problematic.
    - First, was the baseline trained using the same curriculum-based approach as CoEvolve?
        - If not, RL policies are trained directly on difficult environments from the start, whereas CoEvolve uses code-as-policy (CAP) with a curriculum.
        - This makes the comparison unfair, and it is invalid to conclude that CAP outperforms the RL policy.
    - The learning curves show high initial rewards that decrease over training.
        - While the reward return can be truncated in RL, a decline in performance raises serious concerns that the hyper-parameters may have been improperly tuned or the experiment misconfigured.

---

### Official Review · Reviewer_pTWM · 2025-11-01

**Soundness:** 2
**Presentation:** 2
**Contribution:** 2
**Rating:** 2
**Confidence:** 3

**Summary:**

This paper proposes to co-evolve agents and environments, represented as code, by framing things as a minimax game between an environment designer and a policy generator. They use PSRO as the self-play algorithm for environment and generator.

**Strengths:**

I think this is an interesting approach, of generating both environments and code, while using PSRO to find the equilibrium of this selection process. It also has nice results indicating that without the curriculum the final levels would be too hard. Finally, the generated levels seem to be quite diverse and interesting.

**Weaknesses:**

- I somewhat object to this sentence in the intro "However, UED typically generates environments via randomization or simple heuristics – rather than intelligently – limiting diversity and relevance." This is a pretty strong, and I think false, statement given that many UED methods use learning algorithms in the loop to try to design the environments. Also, just as a rule, calling your method "intelligent" and other methods as "unintelligent" is uh, not great. Relatedly, the paper contains claims like "Crucially, our LLM-driven co-evolution introduces intelligent, data-driven priors that enable the design of more challenging and relevant environments than classical, heuristic-based UED" but does not contain comparisons to methods from the UED literature, just to eurekaverse.

- The biggest issue in this paper is that there's an asymmetry between the generator and the policy wherein the generator can just create totally unsolvable environments. The appendix vaguely mentions that they only include levels if they're solvable but this needs to be way more prominent if this is the case because it's very crucial! Without this, the fact that it works at all would entirely be contingent on the weakness of the LLM as a generator i.e. it is unable to find these unsolvable maps. The very first time an LLM generates an unsolvable map, this will be the only sampled map in the future. I'm assuming that the point in the appendix about only including a map if some policy solves it is how this is resolved but this needs to be much more prominently displayed.

- The main change in this paper, as far as I can tell, is to use PSRO instead of simply keeping the next best policy as in their eurekaverse policy. The subsequent improvement this entails seems to be quite small, though, it is hard to tell because they use only a single run.

**Questions:**

- I'm not sure I understand the logic, or soundness, of reporting results from a single run. The claim in the paper is that this would mess things up since evaluation sets would be different across runs. The solution would seem to be to design your evaluation set in advance. Have I perhaps misunderstood your statistical estimation procedure? In table 1, you report a standard error (I think?) is this across seeds or rollouts?
- Why do the scores of the policies decrease across levels in figure 6?

---

### Official Review · Reviewer_1ToJ · 2025-11-01

**Soundness:** 2
**Presentation:** 3
**Contribution:** 3
**Rating:** 4
**Confidence:** 3

**Summary:**

The paper introduces Covolve, a framework for co-evolving LLM-generated policies and environments. Each environment and policy is expressed as Python code produced by a large language model. Covolve formulates this interaction as a two-player zero-sum game between an environment designer and a policy designer. The framework maintains populations of both, evaluates their pairwise interactions to form a payoff matrix, and computes a mixed-strategy Nash equilibrium using a Policy Space Response Oracle procedure. The MSNE is then used to generate new environments adversarially and new policies as best responses, forming an automated curriculum. Experiments are presented on MiniGrid, PyGame 2D navigation, and CARLA driving. Reported results include visual examples of environments that become more complex over iterations, performance comparisons against the Eurekaverse baseline, ablations showing the effect of MSNE mixtures versus latest-only updates, and limited generalization tests on a few standardized environments.

**Strengths:**

- The setup as a two-player zero-sum game with empirical PSRO computation is explicitly defined and implemented.
- Policies and environments are represented as Python code that can be directly executed, improving transparency relative to neural-only representations.
- The appendix includes pseudocode, LLM prompts, and environment specifications.
- Figures show increasing environment complexity over iterations.
- Plots comparing Covolve’s MSNE mixture against Eurekaverse’s latest-only policy show that performance on earlier environments decreases less sharply.
- Positive generalization tests.

**Weaknesses:**

- The paper states that runs diverge because generated environments differ. All main results come from a single run per domain. There are no averages or standard deviations across random seeds, limiting statistical confidence.
- Generalization is tested on three environments only (two MiniGrid, one CARLA). No results are reported on PyGame test environments outside the training loop.
- The paper notes that generated levels may be unsolvable and that helper functions or reachability checks are used to filter them. The frequency or effect of these filters is not reported.
- Eurekaverse is the only adaptive baseline evaluated. Other unsupervised environment design methods (PAIRED, ACCEL, ...) are mentioned but not compared experimentally.
- The framework evaluates multiple LLM-generated policies and environments per iteration, but runtime, computational cost, and scaling with candidate count are not reported.
- The MSNE equilibrium is computed on a finite payoff matrix built from a limited archive of policies and environments. Robustness is shown only within this archive and extrapolation to unseen tasks is not analyzed beyond Table 1.
- Success rates in tables are given as means but not accompanied by confidence intervals or number of trials, limiting interpretability.

**Questions:**

- What proportion of generated environments fail solvability or reachability checks during evolution?
- How many iterations and candidates per iteration are used, and what are the associated compute costs?
- Are the generalization results stable across multiple independent runs?
- How sensitive is the MSNE equilibrium to population size and candidate sampling?
- Could comparisons to existing UED methods (PAIRED, ...) be added under similar compute budgets?

---

### Author Response · Authors · 2025-12-03

We thank the reviewers for their feedback. We are withdrawing this submission and will resubmit after strengthening the work. For clarity, we note several methodological details that were not fully visible in the current draft: (i) environment solvability is enforced before a generated environment is added to the set of evaluated environments, so unsolvable cases do not influence the policy–environment payoff matrix; (ii) the Eurekaverse baseline used here is a latest-only variant adapted to code-as-policy and does not use MSNE; and (iii) reporting based on a single run can hide variability, and additional independent runs will make performance differences clearer. We will also provide clearer definitions of how evaluation statistics are computed and how computational cost scales with the number of evaluated policies and environments.

---

### Note · Authors · 2026-01-22

I have read and agree with the venue's withdrawal policy on behalf of myself and my co-authors.